# Oxidation of DJ-1 Cysteines in Retinal Pigment Epithelium Function

**DOI:** 10.3390/ijms23179938

**Published:** 2022-09-01

**Authors:** Sanghamitra Bhattacharyya, Johnathon Sturgis, Arvydas Maminishkis, Sheldon S. Miller, Vera L. Bonilha

**Affiliations:** 1Cole Eye Institute, Cleveland Clinic, Cleveland, OH 44195, USA; 2Department of Molecular Medicine, Cleveland Clinic Lerner College of Medicine, Case Western Reserve University, Cleveland, OH 44195, USA; 3National Eye Institute, National Institutes of Health, Section on Epithelial and Retinal Physiology and Disease, Bethesda, MD 20892, USA; 4Department of Ophthalmology, Cleveland Clinic Lerner College of Medicine, Case Western Reserve University, Cleveland, OH 44195, USA

**Keywords:** retinal pigment epithelium, retinal degeneration, oxidative stress, DJ-1, cysteine

## Abstract

The retina and RPE cells are regularly exposed to chronic oxidative stress as a tissue with high metabolic demand and ROS generation. DJ-1 is a multifunctional protein in the retina and RPE that has been shown to protect cells from oxidative stress in several cell types robustly. Oxidation of DJ-1 cysteine (C) residues is important for its function under oxidative conditions. The present study was conducted to analyze the impact of DJ-1 expression changes and oxidation of its C residues on RPE function. Monolayers of the ARPE-19 cell line and primary human fetal RPE (hfRPE) cultures were infected with replication-deficient adenoviruses to investigate the effects of increased levels of DJ-1 in these monolayers. Adenoviruses carried the full-length human DJ-1 cDNA (hDJ) and mutant constructs of DJ-1, which had all or each of its three C residues individually mutated to serine (S). Alternatively, endogenous DJ-1 levels were decreased by transfection and transduction with shPARK7 lentivirus. These monolayers were then assayed under baseline and low oxidative stress conditions. The results were analyzed by immunofluorescence, Western blot, RT-PCR, mitochondrial membrane potential, and viability assays. We determined that decreased levels of endogenous DJ-1 levels resulted in increased levels of ROS. Furthermore, we observed morphological changes in the mitochondria structure of all the RPE monolayers transduced with all the DJ-1 constructs. The mitochondrial membrane potential of ARPE-19 monolayers overexpressing all DJ-1 constructs displayed a significant decrease, while hfRPE monolayers only displayed a significant decrease in their ΔΨm when overexpressing the C2S mutation. Viability significantly decreased in ARPE-19 cells transduced with the C53S construct. Our data suggest that the oxidation of C53 is crucial for regulating endogenous levels of ROS and viability in RPE cells.

## 1. Introduction

The retinal pigment epithelium (RPE) is a post-mitotic monolayer comprised of polarized, pigmented cells that separate the choroidal and photoreceptor of the distal retina. Due to its location, the RPE tight junctions function as a fundamentally important component of the blood–retina barrier. By orchestrating the movement of metabolites, water, and ions between the retina and the choriocapillaris, the RPE monolayer also assists in maintaining the health and integrity of the retina and the transfer of visual information to the brain [1,2,3].

Reactive oxygen species (ROS) are derivatives of partially reduced molecular oxygen that are formed and degraded by all multicellular aerobic organisms during metabolic processes by enzymes, such as nicotinamide adenine dinucleotide phosphate [NADPH] oxidases (Noxes), other oxidases, lipoxygenases, and other components of the electron transport chain [4]. Moderate amounts of ROS are beneficial to the cell, whereas high or low amounts make the cell susceptible to oxidative stress. Thus, ROS is an active regulator of cellular signaling and is balanced by robust antioxidative systems under physiological conditions [5]. Antioxidants are molecules capable of maintaining cellular function due to their ability to stabilize or deactivate the ROS, thereby protecting the cells against auto-oxidative damage [6]. Oxidative stress has been considered a factor in RPE and retinal degeneration leading to pathophysiological outcomes. Oxidative stress occurs due to increased endogenous and exogenous ROS generation and decreased antioxidative capability; it can lead to metabolic dysfunction, cell senescence, and cell death [7]. As a tissue with high metabolic demand and ROS generation, the retina and RPE cells are regularly exposed to chronic oxidative stress. However, these cells contain mechanisms to regulate ROS generation endogenously.

DJ-1 is a chaperone protein that functions as a redox-sensitive molecule protecting cells from oxidative damage. It is universally expressed in various tissues, including the brain and the retina [8]. DJ-1 levels are increased in cells under an oxidative stress condition [9,10,11]. Moreover, DJ-1 overexpression protects against oxidative injury, whereas knockdown of DJ-1 increases susceptibility to oxidative damage [12,13]. These deleterious effects of DJ-1 reductions/loss can be mitigated by restoring DJ-1 expression.

Crystal structure analysis of the purified DJ-1 protein has identified three cysteine (C) residues at amino acid numbers 46, 53, and 106 (C46, C53, and C106) in humans and rats [14,15]. DJ-1 functions as a ROS scavenger protein due to its ability to quench ROS by self-oxidation of its cysteine residues [14,16,17,18,19]. However, high oxidation leads to the inactivation of DJ-1 function due to irreversible C-SO_3_H formation, resulting in severe disease progression [20,21]. Previous studies carried out on various types of cultured mammalian cells identified C106 as the critical residue for DJ-1-mediated protection against oxidative stress due to sequential oxidation to form sulfenic acid (SOH), sulfinic acid (SO_2_H), and sulfonic acid (SO_3_H) [14,22,23,24,25]. The peripheral cysteine residues, C46 and C53, are present at the dimeric interface of DJ-1 and are important for DJ-1 cytoprotective function mediated by ASK1 binding [15,26,27,28]. In addition, DJ-1 is also susceptible to other post-translational modifications (PTMs), such as sumoylation [29], S-nitrosylation [30,31], and phosphorylation [32,33]. Here, we analyzed the function of DJ-1 cysteine oxidation in RPE cells with different levels of endogenous DJ-1. We demonstrated that RPE cells with decreased levels of DJ-1 (ARPE-19 + shDJ-1 cells) are more vulnerable to low oxidative stress. Localization experiments determined that overexpression of full-length DJ-1 and C106S results in increased distribution to the RPE mitochondria. Finally, we also demonstrated that oxidation of C53 is critical for DJ-1 to exert its antioxidant activity in RPE cells. Altogether, these results suggest that DJ-1 undergoes unique PTMs in RPE cells.

The present study analyzed the extensively studied and largely known ARPE-19 cell line and primary hfRPE cells well known to closely mimic the function of native human tissue [34]. Furthermore, primary hfRPE cultures have been extensively investigated, and the data obtained were crucial for developing the current use of iPSC-induced human RPE for therapeutic intervention in several neurodegenerative RPE diseases [35,36,37,38,39]. Thus, the present study adds support for cell therapy as a potentially valuable strategy to treat AMD patients by directly replenishing degenerated RPE via autologous transplantation of induced pluripotent stem-cell-derived RPE. The latter approach is the basis of an ongoing clinical trial utilizing autologous transplantation of induced pluripotent stem-cell-derived RPE in patients with geographic atrophy (NCT04339764).

## 2. Results

### 2.1. Downregulation of DJ-1 in ARPE-19 Cells Results in Increased Levels of Oxidative Stress

Our previous studies reported that DJ-1 provides a critical antioxidant defense against oxidative stress mediated by multiple molecular processes, including increased DJ-1 levels, increased expression of several antioxidant genes, and oxidation of conserved DJ-1 cysteine residues, to scavenge excess ROS [11,40]. To test the role of DJ-1 levels in RPE monolayers, we knocked down DJ-1 mRNA and protein levels by transfection of RPE cells with five predesigned MISSION^®^ TRC shRNA constructs specific to human PARK7 gene and control shRNA using Lipofectamine^®^ 2000 (Appendix A). Stable cell clones were established for each predesigned PARK7 shRNA, and Western blots determined the DJ-1 downregulation efficiency (Appendix A). The Western blot analysis revealed that the DJ-1 levels were visibly decreased in cells transfected with all the predesigned PARK7 shRNA (Appendix A, lanes 2 to 4). However, ARPE-19 cells transfected with PARK7 shRNA NM_007262.3-506s1c1 (clone #4, Appendix A, lane 4) displayed the most visibly significant decreased levels of DJ-1. Thus, we proceeded with further experiments using this clone (referred to as ARPE-19 + shDJ-1) and compared our findings in this clone to untransfected ARPE-19 cells.

Monolayers of ARPE-19 and clones of ARPE-19 cells transfected with no shRNA (ARPE-19 + shctrl) and PARK7 shRNA (ARPE-19 + shDJ-1) were obtained. The initial characterization of these three RPE cell lines analyzed the expression of DJ-1 by Western analysis (Figure 1A,B). A major band of ~25 kDa was observed in the extracts of all the RPE cell lines (Figure 1A, lanes 1 to 3). Immunoblots of RPE lysates obtained from all RPE monolayers demonstrated significant knockdown of DJ-1 in the ARPE + shDJ-1 clone compared to the levels of ARPE-19 and ARPE-19 + shctrl clone. Quantitation of the intensity of DJ-1 immunoreactivity in blots detected a four-fold decrease in ARPE-19 + shDJ-1 clone compared to ARPE-19 monolayers and three-fold compared to ARPE-19 + shctrl (Figure 1B). Exposure of proteins to oxidative stress results in oxidation of protein side chains and generation of carbonyl derivatives [41]. Thus, we also analyzed and measured the levels of carbonyl groups in the protein side chains after derivatization to 2,4-dinitrophenylhydrazone (DNP-hydrazone) by reaction with 2,4-dinitrophenylhydrazine (DNPH) in ARPE-19 monolayers and the ARPE-19 + shDJ-1 and ARPE-19 + shctrl clones (Figure 1C). Immunoblots of RPE lysates obtained from all RPE monolayers demonstrated a significant increase in DNP levels in the ARPE + shDJ-1 clone compared to the levels of ARPE-19 and ARPE-19 + shctrl clone. Quantitation of the intensity of DNP immunoreactivity in blots from three independent experiments detected a two-fold increase in ARPE-19 + shDJ-1 clone compared to ARPE-19 monolayers and ARPE-19 + shctrl (Figure 1D).

### 2.2. Expression of Exogenous DJ-1 and DJ-1 Cysteine Mutants in ARPE-19 Polarized Monolayers with Different Levels of Endogenous DJ-1 Affects Its Response to Oxidative Stress

Adenovirus-mediated gene delivery has been an essential tool for assessing the functional competence of the molecule delivered into the cell. Previous reports showed that the cytoprotective activity of DJ-1 against oxidative stress depends on its cysteine residues [17,23]. Thus, in this context, ARPE-19 and ARPE-19 +shDJ-1 cells were transduced with control adenoviruses carrying an empty vector (Ad) and the full-length human DJ-1 cDNA (hDJ-1); monolayers were also transduced with four different DJ-1 mutants: i—a construct that has the C residues at amino acids 46, 53, and 106 mutated to serine (S) (C2S); ii—a construct that has the C residues at amino acids 46 mutated to S (C46S); iii—a construct that has the C residues at amino acids 53 S (C53S); iv—a construct that has the C residues at amino acids 106 mutated to S (C106S). We performed a PCR reaction with primers specific to the CMV promoter present in all the adenovirus constructs to determine the viral expression levels in the infected monolayers. Immunofluorescence using an adenovirus type 5 antibody was also performed to visualize the viral expression in the monolayers (Appendix A). Quantification of the CMV promoter in the transduced ARPE-19 monolayers revealed similar overexpression levels between the different adenovirus constructs (Appendix A). Confocal microscopy of the ARPE-19 monolayers with an antibody to adenovirus type 5 revealed a similar expression pattern of the viral proteins in all the samples, irrespective of the harboring gene sequence (Appendix A).

Confocal microscopy *en face* examination of ARPE-19 (Figure 2A–F,M–R) and ARPE-19+ shDJ-1 (Figure 2G–L,S–X) monolayers grown on polycarbonate filters, fixed, and reacted with an antibody to DJ-1 was also performed. Previous experiments carried out in our lab reported a dose–response relationship between levels of DJ-1 and oxidized DJ-1 intensity in RPE monolayers exposed for 1 and 18 h to increasing concentrations of H_2_O_2_ [11]. Those experiments also identified a visible increase in immunocytochemical staining for DJ-1 under these conditions and established that overnight incubation with 400–1000 μM H2O2 resulted in significant RPE cell death. Thus, in this study, we incubated RPE monolayers with 200 μM H_2_O_2_ for 18 h as a model of low chronic oxidative stress.

A qualitative comparison between ARPE-19 and ARPE-19 + shDJ-1 displayed visually significant downregulation of DJ-1 in the ARPE-19 + shDJ-1 monolayers both under baseline conditions (Figure 2G) and in conditions of low chronic oxidative stress (Figure 2S) when compared to ARPE-19 monolayers (Figure 2A,M). Moreover, under baseline conditions, both monolayers transduced with the C2S, C46S, and C53S constructs displayed visually lower immunoreactivity for DJ-1 (Figure 2C–E,I–K). Confocal microscopy also revealed that, in both monolayers under baseline conditions, DJ-1 displays a diffuse cytoplasmic pattern, as previously reported [11]. In ARPE-19 monolayers transduced with the Ad, hDJ-1, C46S, and C106S constructs, some cells also displayed nuclear staining at baseline conditions (Figure 2A,B,F, arrows) and in conditions of oxidative stress (Figure 2M,N,R, arrows). ARPE-19 + shDJ-1 monolayers transduced with all the DJ-1 constructs failed to display DJ-1 nuclear staining in any condition.

Our previous studies determined that exposure of ARPE-19 cells to oxidative stress was sufficient to enhance DJ-1 levels [11]. Increased labeling for DJ-1 was observed in both monolayers transduced with the DJ-1 constructs and exposed to oxidative stress, except for ARPE-19 + shDJ-1 transduced with the control Ad (Figure 2S) and C46S (Figure 2V) adenoviruses. Quantitation of DJ-1 immunoreactivity was carried out in blots of lysates from parallel samples (Figure 3). Significant increased DJ-1 levels (>1.5-fold increase) were observed in all RPE monolayers transduced with the DJ-1 constructs; higher levels were observed in monolayers transduced with the hDJ-1 and C46S adenoviruses (Figure 3A,C). Interestingly only ARPE-19 monolayers transduced with Ad construct and incubated with 200 μM H_2_O_2_ were able to display a significant increase of 1.3 fold in DJ-1 levels when compared with ARPE-19 transduced with Ad under baseline conditions (Figure 3A,C); ARPE-19 transduced with all the other DJ-1 constructs did not significantly increase their DJ-1 levels. Under baseline conditions, ARPE-19 + shDJ-1 displayed significant downregulation (0.7-fold) of DJ-1 levels in the monolayers transduced with the control Ad vector. In comparison, the intensity of DJ-1 immunoreactivity was 2.6- and two-fold higher in monolayers transduced with hDJ-1 and C106S constructs (Figure 3B,D). Exposure of ARPE-19 + shDJ-1 monolayers to low chronic oxidative stress resulted in 3.8-fold increase in DJ-1 immunoreactivity in monolayers transduced with the hDJ-1 adenovirus, the most significant upregulation of DJ-1 in this monolayer (Figure 3B,D). Moreover, the data obtained suggested that infection with the adenoviruses potentially stressed the monolayers and increased endogenous DJ-1 levels.

Additional data analysis identified an apparent inconsistency between the experiments analyzing the distribution of DJ-1 and quantifying the overall levels of DJ-I signal in ARPE-19. ARPE-19 + shDJ-1 monolayers overexpressed the different DJ-1 constructs under baseline and oxidative stress conditions (Figure 2 and Figure 3). Examples included the lower intensity labeling for DJ-1 of monolayers transduced with DJ-1 constructs C2S, C46S, and C53S compared to monolayers transduced with the empty adenovirus construct by confocal microscopy. However, the DJ-1 levels in lysates of monolayers transduced with the DJ-1 constructs C2S, C46S, and C53S were higher than monolayers transduced with the empty adenovirus construct by Western blot. These differences could be related to the different affinity of the DJ-1 antibodies used in each analysis as the monoclonal used in the confocal analysis was generated by immunization with recombinant full-length DJ-1. In contrast, the polyclonal used in the Western blot analysis was generated by immunization with C-terminus (residues 150–189) of human DJ-1. It is also important to remember that the confocal analysis was carried out after fixation and extraction of the monolayers. Thus, it may be indicative of differences in the solubility and increased cytoplasmic localization of the exogenous DJ-1 protein within the cell. Moreover, it is important to remember that confocal microscopy is not a quantitative technique and the panel images were adjusted for brightness and contrast to highlight detected proteins.

### 2.3. Expression of Exogenous DJ-1 and DJ-1 Cysteine Mutants in Polarized RPE Monolayers with Different Levels of Endogenous DJ-1 Affects Mitochondria Structure and Function

Our previous studies reported that more DJ-1 was localized to the mitochondria under oxidative stress conditions [11]. Therefore, to test the functional consequences of oxidation of DJ-1 cysteines in RPE monolayers, we transduced the monolayers with DJ-1 adenoviruses constructs and infected RPE monolayers with CellLight^®^ Mito-GFP to evaluate the mitochondrial morphology. The monolayers were then counterstained with an anti-adenovirus antibody to detect successfully infected cells capable of expressing exogenous DJ-1 as a tag was not engineered into these constructs. Analysis was conducted under baseline (Figure 4) and low chronic oxidative stress (Figure 5) conditions.

At baseline conditions, confocal microscopy of ARPE-19 monolayers transduced with Ad control construct (Figure 4A,M) displayed highly elongated and interconnected mitochondria. Transduction with all the DJ-1 constructs resulted in morphological changes in the mitochondria that expanded the mitochondrial network and interconnectivity in the RPE cytoplasm, together with the appearance of dilated tubules (Figure 4A–R, arrows). However, the ARPE-19 monolayers transduced with the C53S construct displayed more fragmented and shorter tubular structures (Figure 4E,Q). ARPE-19 monolayers transduced with the control Ad displayed little colocalization between exogenous DJ-1 (labeled with adenovirus antibody) and mitochondria (Figure 4G,M), suggesting that, under baseline conditions, the majority of DJ-1 is not present in the mitochondria of the RPE cells as previously reported. However, transduction with exogenous DJ-1 induced increased localization of DJ-1 to mitochondria in ARPE-19 monolayers transduced with the hDJ-1 (Figure 4H,N, arrowheads), C46S (Figure 4J,P, arrowheads), and C106S (Figure 4L,R, arrowheads) constructs. ARPE-19 + shDJ-1 monolayers transduced with the Ad control construct (Figure 4A’,M’, double arrows) displayed smaller and shorter mitochondria when compared to ARPE-19 monolayers (Figure 4A,M). ARPE-19 + shDJ-1 monolayers transduced with hDJ-1 (Figure 4B’,N’) and the C46S (Figure 4D’,P’) construct displayed a more elaborated mitochondrial network, but these were more confined to the center of the RPE cells. The mitochondria were very fragmented in ARPE-19 + shDJ-1 monolayers transduced with the C2S construct (Figure 4C’,O’). However, they were highly elongated and interconnected in monolayers transduced with the C53S (Figure 4E’,Q’) and C106S (Figure 4F’,R’) constructs.

In support of our findings with the ARPE-19 monolayers, we have also analyzed hfRPE monolayers transduced with some of the same DJ-1 constructs followed by infection with CellLight Mito-GFP to evaluate the mitochondrial morphology (Figure 5). Confocal microscopy of hfRPE transduced with the control Ad construct displayed elongated mitochondria and highly connected tubules extended through the entire RPE cell cytoplasm (Figure 5A). However, dilated tubules were observed in hfRPE cultures transduced with both the hDJ-1 (Figure 5B, arrows) and C2S constructs (Figure 5C, arrows). Conversely, transduction of the hfRPE monolayer with a lentivirus carrying the human short hairpin (sh)RNA DJ-1 previously used to generate the ARPE-19 + shDJ-1 clone displayed an increased number of thin over-extended tubules (Figure 5D). Immunoblots lysates of parallel hfRPE monolayers confirmed that cultures transduced with hDJ-1 (Figure 5E, lane 3) and with the C2S Ad constructs (Figure 5E, lane 2) displayed significantly increased immunoreactivity of DJ-1 when compared to control cultures (Figure 5E, lane 1) and normalized to the levels of GAPDH. Conversely, infection of hfRPE monolayers with shRNA DJ-1 lentivirus significantly decreased DJ-1 immunoreactivity (Figure 5E, lane 4). Quantitation of the intensity of immunoreactivity in blots showed that DJ-1 increased 2.1- and 2.8-fold in the hfRPE monolayers overexpressing hDJ-1 and C2S constructs but decreased 0.3-fold in the hfRPE monolayers infected with the shRNA DJ-1 lentivirus compared with control cell RPE cultures (Figure 5F).

At low oxidative stress conditions, confocal microscopy of ARPE-19 monolayers transduced with the Ad control (Figure 6A,M), hDJ-1 (Figure 6B,N), C53S (Figure 6E,Q), and C106S (Figure 6F,R) constructs displayed highly elongated and fragmented mitochondria. Mitochondria with dilated tubules were not observed under this condition in ARPE-19 monolayers transduced with any DJ-1 construct. Moreover, ARPE-19 monolayers transduced with the C2S (Figure 6C,O) and C46S (Figure 6D,P) constructs displayed mitochondria more concentrated in the center of the RPE cells. ARPE-19 (Figure 6H,N, arrowheads) and ARPE-19 + shDJ-1 (Figure 6H’,N’, arrowheads) monolayers transduced with the hDJ-1 construct displayed highly specific colocalization between the exogenous DJ-1 labeled with adenovirus antibody and mitochondria. ARPE-19 + shDJ-1 monolayers transduced with the Ad control (Figure 6A’,M’), hDJ-1 (Figure 6B’,N’), C2S (Figure 6C’,O’), and C46S (Figure 6D’,P’) constructs displayed smaller and shorter mitochondria when compared to ARPE-19 monolayers (Figure 6A,M). Mitochondria with dilated tubules were present in monolayers transduced with the hDJ-1 (Figure 6B’,N’, arrows), C2S (Figure 6C’,O’, arrows), C46S (Figure 6D’,P’, arrows), and C106S (Figure 6F’,R’, arrows) constructs. ARPE-19 + shDJ-1 monolayers transduced with the other hDJ-1 (Figure 6B’,N’) and C46S (Figure 6D’,P’) displayed mitochondria more concentrated in the center of the RPE cells. The mitochondria in ARPE-19 + shDJ-1 monolayers transduced with C53S (Figure 6E’,Q’) and C106S constructs (Figure 6F’,R’) were highly elongated and interconnected.

### 2.4. Expression of Exogenous DJ-1 and DJ-1 Cysteine Mutants in ARPE-19 Polarized Monolayers with Different Levels of Endogenous DJ-1 Affects RPE Response to Oxidative Stress

The cysteine residues of DJ-1 play an important role in its biological function due to accumulating evidence suggesting that it responds to oxidative stress by oxidizing its cysteine residues. To test whether DJ-1 cysteine residues affect RPE cell viability, the cells were incubated with low chronic H_2_O_2_ for 17 h, after which mitochondrial membrane potential (ΔΨm) and viability were assessed (Figure 7). Monolayers were incubated with tetramethylrhodamine ethyl ester (TMRE), and the fluorescence signal was acquired to determine the mitochondrial membrane potential (ΔΨm) (Figure 7A–C). The ARPE-19 monolayers overexpressing all DJ-1 constructs displayed a significant decrease in their ΔΨm (Figure 7A), while the hfRPE monolayers displayed a significant decrease in their ΔΨm when overexpressing C2S-mutated DJ-1 (Figure 7C). Alternatively, the ARPE-19 + shDJ-1 monolayers displayed a trend of increased ΔΨm in monolayers overexpressing all but the C2S DJ-1 constructs; significantly increased ΔΨm was detected in monolayers overexpressing the full-length DJ-1 and the C106S mutant constructs (Figure 7B). The viability was significantly decreased in ARPE-19 cells transduced with the C53S and C106S constructs in both the baseline and oxidative stress conditions. In ARPE-19 + shDJ-1, transduction with these constructs also significantly decreased the viability both in the baseline and conditions of oxidative stress in comparison to ARPE-19 cells (Figure 7D).

## 3. Discussion

Oxidative stress occurs when there is excessive production of metabolic byproducts of reduction–oxidation reactions, such as ROS, or insufficient antioxidant activity of free radicals in all living cells. These highly reactive forms must be scavenged and rapidly metabolized by exogenous or endogenous antioxidant systems to keep their level below a critical threshold [42]. In the eye, oxidative stress has been linked to the pathogenesis of lens cataracts, glaucoma, diabetic retinopathy, retinitis pigmentosa, Stargardt macular dystrophy, Sorsby fundus dystrophy, and age-related macular degeneration [43,44,45,46,47,48,49,50,51]. DJ-1 is a multifunctional protein that plays a cytoprotective role and counteracts oxidative stress in neurons [17,52]. The antioxidant properties of DJ-1 have been documented in various neurodegenerative ailments, such as amyotrophic lateral sclerosis, Parkinson’s, Alzheimer’s, and Huntington’s disease. The levels of DJ-1 have also been shown, for the first time, to be increased in RPE cells in vitro [11], as well as in the retinas and RPE isolated from human donors and from aging and stressed mice [53,54]. Specifically, we reported that the absence of DJ-1 accelerates aging in DJ-1 KO mice retinas and RPE, increases their susceptibility to oxidative damage, and induces degeneration even under conditions of low oxidative stress and independent of age [40]. Here, we analyzed the role of DJ-1 cysteine oxidation in RPE homeostasis at the baseline and in conditions of low oxidative stress. A summary of the changes identified is shown in Table 1.

Our study with the DJ-1 knockdown RPE cell line (ARPE-19 + shDJ-1) provides a better understanding of DJ-1 function in RPE cells with normal endogenous levels of DJ-1 (Figure 1). The data agree with previous studies reporting that the downregulation of DJ-1 resulted in decreased cell viability under conditions of oxidative stress. At the same time, under baseline conditions, it increased ROS levels in cardiac microvascular endothelial cells, corneal endothelial cells, and renal tubular epithelial cells [55,56,57].

Our previous studies determined that exposure of ARPE-19 cells to oxidative stress was sufficient to enhance DJ-1 levels [10]. Here, we showed that overall levels of DJ-1 were not significantly enhanced in ARPE-19 monolayers transduced with all the DJ-1 constructs and exposed to oxidative stress (Figure 3). This observation suggests a limit in the levels of expression of DJ-1 by RPE cells. As such, maximal levels of DJ-1 might have already been reached by the overexpression of the exogenous DJ-1 constructs. However, in ARPE-19 + shDJ-1 monolayers, the full-length DJ-1 was the only construct significantly upregulated in ARPE-19 monolayers exposed to low levels of oxidative stress. Our results suggest that oxidation of all the DJ-1 cysteines is needed for the upregulation of the protein in response to oxidative stress.

DJ-1 localizes to the cytoplasm, nucleus, and mitochondria, where it exerts specific cytoprotective functions in each of these compartments [15,58]. Some cells displayed nuclear staining at baseline and oxidative stress conditions in ARPE-19 monolayers transduced with the Ad, hDJ-1, C46S, and C106S constructs (Figure 2). Nuclei localization was not observed in the ARPE-19 + shDJ-1 monolayers. Previous studies showed nuclear translocation of DJ-1 in neuronal cells exposed to oxidative stress as part of the cytoprotective role of DJ-1 [59,60]. DJ-1/PARK7 does not exhibit any distinct DNA-binding domains, suggesting it likely acts as a co-activator that regulates the activity of transcription factors, such as p53, polypyrimidine tract-binding protein-associated splicing factor (PSF), KEAP-1, and NRF-2, among others. Therefore, the loss of nuclear localization observed here is likely to significantly affect the antioxidant role of DJ-1. In the ARPE-19 monolayers, there was a mixture of endogenous and exogenous DJ-1 proteins, while, in the ARPE-19 + shDJ-1 monolayers, most DJ-1 was exogenous. Thus, in the ARPE-19 monolayer, endogenous DJ-1 might mask the overexpression of the exogenous DJ-1 constructs. Overall, our results suggest that oxidation of DJ-1 C53 is needed for nuclei localization in RPE cells under oxidative stress conditions.

Interestingly, confocal microscopy of hfRPE transduced with a lentivirus carrying the human PARK7 short hairpin (sh)RNA, previously used to generate the ARPE-19 + shDJ-1 clone, displayed an increased number of thin over-extended tubules (Figure 5D). The observed differences could be because viral transduction using a lentivirus vector is a far more efficient and successful way of expressing a gene of interest than simple lentivirus plasmid transfection. Alternatively, the differences could be related to the methodology used for the down-expression of DJ-1 since hfRPE monolayers were transiently transduced with the shDJ-1 lentivirus. At the same time, the ARPE-19 + shDJ-1 cell line was transfected, cells were selected, and a clone containing transfected DNA that has integrated into the cellular genome was isolated after antibiotic selection.

Oxidation of C106 is important but not sufficient for mitochondrial localization of DJ-1 [23,61,62]. Specifically, mutation of C106 to alanine (C106A) and serine (C106S) DJ-1 constructs is still localized to the mitochondria [62,63]. Altogether, these results indicate that mitochondrial localization of DJ-1 is enhanced by Cys106 oxidation, although oxidation of this residue is not a requirement for DJ-1 to associate with the mitochondria. In ARPE-19 and hfRPE, the overall levels of DJ-1 affected mitochondria structure and morphology, as analyzed by the appearance of dilated tubules (Figure 4 and Figure 5) in each RPE monolayer following transduction with all DJ-1 constructs. In addition, ARPE-19+ shDJ-1 monolayers displayed smaller and shorter mitochondria, associated with decreased levels of DJ-1. In ARPE-19 monolayers, transduction with the hDJ-1, C46S, and C106S constructs increased mitochondrial localization of the exogenous DJ-1 labeled with adenovirus antibody. However, no significant mitochondrial localization of the exogenous DJ-1 was observed in the ARPE-19 + shDJ-1 monolayers, suggesting that, although C46 and C106 may be important for mitochondrial localization, the presence of some endogenous full-length DJ-1 is important for the mitochondrial localization in the ARPE-19 monolayers. Low oxidative stress conditions promoted DJ-1 translocation to the mitochondria in ARPE-19 and ARPE-19 + shDJ-1 monolayers transduced with the hDJ-1 construct. Our observations suggest that, in RPE cells, all cysteine residues need to become oxidized in response to oxidative stress and promote DJ-1 translocation into the mitochondria of these cells. However, these responses may be modified according to the intensity and nature of the oxidative stress conditions.

RPE metabolism relies on a high density of mitochondria that ensures the elevated metabolic activity of the cells and generates the required number of ATP molecules for accomplishing the physiological functions [64]. Thus, mitochondria function strongly depends on the integrity of its network. Network damage reduces mitochondrial DNA integrity, interchange of mitochondrial material, respiratory capacity, apoptosis, and response to cellular stress, leading to abnormal development and several human diseases [65,66]. Thus, our observations on the RPE mitochondrial network indicate the importance of DJ-1 in maintaining mitochondrial integrity, function, and cell survival in these cells. Our experiments using TMRE labeling (Figure 7) provided additional information about the impact of DJ-1 cysteine residues on RPE health as TMRE is selectively taken up and retained by mitochondria with an intact membrane [67]. Interestingly, we detected decreases in the ΔΨm of both RPE monolayers with endogenous full-length DJ-1, namely the ARPE-19 and hfRPE monolayers. Alternatively, we detected increases in the ΔΨm of RPE monolayers that did not have endogenous full-length DJ-1, namely ARPE-19 + shDJ-1 monolayers. These results suggest that cysteine oxidation of DJ-1 does affect mitochondrial function at baseline conditions. We reason that endogenous DJ-1 might be just enough for proper ΔΨm, but overexpression on top of endogenous levels might overwhelm the cells and drive membrane ΔΨm down. Future experiments will further analyze the role of these in RPE cells under oxidative stress. Moreover, it will also be important to understand how cysteine oxidation of DJ-1 specifically affects mitochondrial ROS and function in baseline and oxidative stress conditions.

Crystal structure analysis of the purified DJ-1 protein has identified three cysteine residues at amino acid numbers 46, 53, and 106 (C46, C53, and C106) in humans and rats [14,15]. Data in the literature suggest that the cytoprotective activity of DJ-1 against oxidative stress depends on the oxidation of these residues. Specifically, it was shown that C106 is the most reactive residue and is both critical for DJ-1 function and very sensitive to oxidative modification [14,23,26]. Reports also showed that C106A and C106S resulted in the loss of DJ-1 cytoprotective activity against oxidative stress and mitochondrial localization [15,23,68,69]. Moreover, it was proposed that DJ-1 C106 is important for the binding of DJ-1 to ASK1 because the mutation of this residue inhibits the protein’s cytoprotective activity and its actual binding to ASK1 (apoptosis signal-regulating kinase 1) [15,70]. In baseline and oxidative stress conditions, viability was significantly decreased in ARPE-19 cells transduced with the C53S and C106S constructs (Figure 7). In ARPE-19 + shDJ-1, transduction with these constructs also significantly decreased viability in baseline and oxidative stress conditions.

C46 and C53 are positioned in the DJ-1 dimer interface and C106 is located on the loop [71]. To our knowledge, only one previously reported study [27] determined that the C53A mutant loses its chaperone activity in response to H_2_O_2_ or protects cells against this oxidative stress agent. Our data suggest that the oxidation of C53 is crucial for regulating endogenous levels of ROS and viability in RPE cells. Alternatively, the data may suggest that dimer formation is crucial for the cytoprotective role of DJ-1 in RPE cells, as well as that C53 may be a potential site associated with DJ-1-mediated protection from chronic stress in the retina. Additional experiments are needed to better understand and further detail the mechanisms of DJ-1 oxidation in RPE function under oxidative stress conditions.

## 4. Materials and Methods

### 4.1. Cell Culture

The human ARPE-19 cell line was obtained from American Type Culture Collection, Gaithersburg, MD, USA, (CRL-2302™, LOT: 70039146). ARPE-19 cells were cultured in DMEM-F12 from Media Core, Cleveland Clinic, Cleveland, OH, USA, (supplemented with 15 mM HEPES, 2.5 mM L glutamine, 0.5% sodium pyruvate, 1200 mg/L sodium bicarbonate, 10% activated fetal bovine serum (FBS), and Penicillin–Streptomycin) at 37 °C with 5% CO_2_. Both WT and DJ-KD cells used here were strictly used within 10 subculture passages.

Primary human fetal RPE (hfRPE) samples were isolated from anonymized donor fetal eyes (16–22 weeks old) as previously reported [34]. Living cells (P0) were shipped in flasks filled with MEM-α modified medium (Sigma-Aldrich, Burlington, MA, USA) supplemented with 5% serum. Confluent monolayer was split once and seeded into Transwells (Corning Costar, New York, NY, USA) coated with human extracellular matrix at 10 μg in 150 μL HBSS per well (BD Biosciences, Franklin Lakes, NJ, USA). For the mitochondria imaging, cells were plated on 35 mm Glass Bottom MatTek Dishes (MatTek Corporation, Ashland, MA, USA).

### 4.2. DJ-1 Knockdown (KD) in RPE Monolayers

ARPE-19 cells were transfected with predesigned MISSION^®^ TRC short hairpin RNA constructs (shRNA) specific to human PARK7 gene (Sigma-Aldrich, Gene ID 11315, TRC_IDs TRCN0000004918 (NM_007262.3-583s1c1), TRCN0000004919 (NM_007262.3-169s1c1), TRCN0000004920 (NM_007262.3-221s1c1), TRCN0000004921 (NM_007262.3-506s1c1)), and control shRNA (Sigma-Aldrich, pLKO.1, SHC001) using Lipofectamine^®^ 2000 according to the manufacturer’s protocol (Invitrogen, Carlsbad, CA, USA, 11668-027). DJ-1 KD clones from each human PARK7 shRNA construct were obtained by puromycin selection (2 μg/mL). Knockdown efficiency was evaluated by Western blot analysis using a commercially available anti-Park7/DJ-1 antibody (Novus Biologicals, Littleton, CO, USA, NB300-270), Appendix A; the clone with the lowest amount of endogenous DJ-1 was selected for experiments described here. DJ-1 KD and ARPE-19 cells were cultured under similar conditions. To promote differentiation, ARPE-19 and DJ-1 KD cells were plated on laminin (Sigma-Aldrich, Burlington, MA, USA, L2020) coated Transwell to the manufacturer’s directions and cultured for 21 days in 1% FBS medium with regular media changes.

Primary hfRPEs were transduced with replication incompetent lentiviral particles, which encode shRNA targeting PARK7 (Sigma-Aldrich, TRCN0000004921). Viral stocks were added at a m.o.i of 2 to the apical surface of the cells in cell culture medium in the presence of hexadimethrine bromide (8 μg/mL) for 14 h at 37 °C. After this incubation, viral particle-containing medium was removed, replaced with fresh medium, and monolayers were returned to the incubator for another 9–11 days.

### 4.3. Adenovirus Infection

All the replication-defective adenovirus vectors Ad5 were prepared and tittered by Welgen Inc. (Worcester, MA, USA) using PARK7 human cDNA clone obtained from Origene (SC115623, Rockville, NY, USA) as previously described [11]. The following constructs were used in this study: CMVPARK7 (for expression of human DJ-1 under control of a human cytomegalovirus [CMV] promoter, hDJ-1), AdCMVPARK7. C to S (C2S, for expression of human DJ-1 with the cysteine at residues 46, 53, and 106 mutated to serine), CMVPARK7C46S (C46S, for expression of human DJ-1 with the cysteine at residues 46 to serine), CMVPARK7C53S (C53S, for expression of human DJ-1 with the cysteine at residues 53 to serine), CMVPARK7C106S (C106S, for expression of human DJ-1 with the cysteine at residues 106 to serine), and the Ad5CMV (Ad, empty vector). To transduce cells, cells were washed with infection medium (20 mM Hepes-buffered DMEM containing 0.2% BSA), and adenovirus stocks were diluted in the same medium and incubated with cells for 3 h at a concentration previously determined to effectively quench ROS when cells were exposed to oxidative stress (100 plaque forming units (pfu)/cell) [11]. Cells on Transwells had viruses added to both apical and basal surface. After infection, infection medium was replaced with medium supplemented with 1% FBS and cells were returned to incubator. Two days after adenovirus transduction, cells were washed with pre-warmed PBS and either fixed with 4% paraformaldehyde made in D-PBS and processed for immunofluorescence or scraped and pelleted down to be processed for biochemistry analysis in 1x RlPA buffer as described below. Alternatively, cells were exposed to oxidative stress one day after adenovirus transduction and then processed for immunofluorescence or biochemistry analysis.

### 4.4. RT-PCR Analysis of Adenoviral Infection

RNA was extracted from each of the adenovirus-infected ARPE-19 monolayers and 500 ng of total RNA was used to make cDNA using oligo dT (Invitrogen). Real Time PCR was carried out using CMV forward 5′-TTCCTACTTGGCAGTACATCTACG-3′ and reverse primer: 5′-GTCAATGGGGTGGAGACTTGG-3′, purchased from IDT. The PCR reaction involved 40 cycles, each with denaturation at 95 °C15 s, annealing at 60 °C/30 s, extension at 72 °C/1 min, and a final extension of 72 °C for 5 min. The cT values for all the Adeno constructs were normalized to gapdh.

### 4.5. Oxidative Stress Conditions and Western Blot Analysis

ARPE-19 and DJ-1 KD ARPE monolayers were seeded on laminin-coated Transwell as described above (Corning, 3419) for 21 days to attain polarity. At day 23, monolayers were infected with adenoviruses as described above. At day 24, adenovirus-infected cells were incubated with 0 and 200 μM H_2_O_2_ (Sigma-Aldrich, H1009) for 17–18 h. At day 25, monolayers were washed in PBS and lysed using 1x RIPA buffer (Pierce, Thermo Scientific, Norristown, PA, USA, 89901) containing protease and phosphatase inhibitors (Sigma-Aldrich, P8340, cocktail2 P5726 and cocktail3 P0044, respectively). The cell pellet with the lysis buffer was vortexed for 30 s and incubated in ice for 10 min and repeated 3 more times. The samples were finally sonicated with a 6 s pulse and centrifuged at 15,000 rpm at 4 °C for 20 min. The supernatant was carefully removed and total protein estimation of the lysates conducted using Pierce BCA Protein Assay Kit (Thermo Scientific, 23227). 30 μg of the total protein was resolved in a 4–20% gradient gel (Biorad, Hercules, CA, USA, 5671094). Protein transfer onto methanol-activated PVDF membranes was carried out at 4 °C maintaining constant voltage of 80 V for 1 h. The membrane was blocked for 2 h with LICOR Intercept buffer and probed for DJ-1 using a rabbit polyclonal anti DJ-1 (Novus, NB300-270, 1:500) or a mouse beta actin (Cell Signaling, Danvers, MA, USA, 8H10D10, 1:4000) overnight at 4 °C, followed by incubation with anti-rabbit IRDye^®^ 800CW or anti-mouse IRDye^®^ 680RD (both from LICOR, Lincoln, NE, USA, 1:10,000). Signal intensities were quantified using ImageJ.

### 4.6. Viability

ARPE-19 and DJ-1 KD ARPE cells were seeded in a 96-well plate at a concentration of 20,000 cells/well and cultured for 6 days. Cells were infected with the various adenovirus on the 7th day and stimulated with 200 μM H_2_O_2_ on the 8th day as described above. Viability was assessed using Presto Blue HS Cell Viability Reagent (ThermoFisher Scientific, P50200) following the manufacturer’s protocol. The fluorescence was measured using VICTOR X2 (PerkinElmer, Waltham, MA, USA) with excitation at 485 nm and emission at 535 nm.

### 4.7. Protein Carbonyl Detection

RPE monolayers (ARPE-19, ARPE-19 + shDJ-1, and ARPE-19 + shctrl) were lysed as described above. Protein carbonyl residues in lysates were quantified in 10 μg of the cell lysates using the Protein Carbonyl Assay Kit (Abcam, Boston, MA, USA, ab178020). Derivatization and neutralization were carried out following the manufacturer’s protocol. A parallel set of lysates were treated with the 1x Derivatization Control solution. Individual oxidized proteins were resolved by SDS-PAGE on 12.5% polyacrylamide gel gradient gel (BioRad, 5671043). Proteins were transferred to PVDF membrane at 70 V for 2–3 h. The membranes were blocked using 5% skimmed milk in PBS for 2 h and incubated with primary anti-DNP rabbit antibody (1:5000) overnight at 4 °C, followed by incubation with anti-rabbit IRDye^®^ 800CW. The same membranes were next incubated with beta actin antibodies as described above. Signal intensity in the derivatized lanes and the control lanes was normalized with the beta actin signals in the respective lanes.

### 4.8. Immunofluorescence of Monolayers

RPE monolayers (ARPE-19 and ARPE-19 + shDJ-1) transduced with the different adenovirus constructs mentioned above and stimulated with 0 and 200 μM H_2_O_2_ were fixed in paraformaldehyde (4%) made in PBS for 20 min at room temperature (RT), followed by quenching in 50 mM ammonium chloride made in PBS for 20 min at RT. Subsequently, monolayers were blocked in PBS + 1% BSA and 0.1% Triton X-100 (Sigma, SLCD3084) for 1 h at RT. Monolayers were incubated with commercially available primary antibodies to: DJ-1 (mouse-Abcam, ab11251, 1:250) and Adenovirus type 5 (rabbit-Abcam, ab6982, 1:500 overnight at 4°C in blocking buffer. Monolayers were incubated with secondary antibodies mouse-Alexa488 (1:1000) and rabbit-Alexa594 (1:1000) for 1 h at RT. Nuclei were stained using TO-PRO^®^-3 iodide (1:10,000; Molecular Probes, Eugene, OR, USA) in PBS for 10 min. Transwells were mounted on microscope slides with Vectashield^®^ mounting medium (Vector Laboratories, Newark, CA, USA, H-1000-10). Monolayers were analyzed using a Leica laser scanning confocal microscope (model TCS-SP8, Wetzlar, Germany). Each individual xy image represents a three-dimensional projection of the entire monolayer (sum of all images in the stack).

### 4.9. Visualization of Mitochondrial Structure in RPE Monolayers Labeled with CellLight^®^ Mitochondria-GFP (Mito-GFP)

Fully confluent RPE monolayers (ARPE-19 and ARPE-19 + shDJ-1) plated for 3 weeks were transduced with CellLight^®^ Mitochondria-GFP, BacMam 2.0 (Invitrogen, C10508) with a particle-per-cell value of 1. Thus, 15 μL BacMam 2.0 reagent was added to 10^5^ cells/well in one well of a laminin-coated 35 mm Glass Bottom MatTek Dishes (MatTek, Corporation, Ashland, MA, USA) following the manufacturer’s protocol. The infection was performed overnight in RPE complete medium after infection with adenovirus as described above. Fully confluent hfRPE monolayers plated for 2 weeks were transduced with CellLight^®^ Mitochondria-GFP, BacMam 2.0 (Invitrogen, C10508) as described above after the adenoviruses and lentiviruses (shPARK7) incubations. Mitochondria morphology was observed on the second day after infection under a confocal microscope (SP8, Leica) after fixation on 4% paraformaldehyde made in D-PBS, blocking, permeabilization of cells, and sequential incubation with an antibody to adenovirus type 5 and an anti-rabbit-Alexa594 antibody. Images were subjected to Lightning Processing (Photon Count) to represent localization more accurately. Microscopic panels were composed using Adobe Photoshop CS3 (Adobe, San Jose, CA, USA).

### 4.10. Tetramethylrhodamine Ethyl Ester Perchlorate (TMRE) Fluorescence Labeling of Monolayers

Fully confluent RPE monolayers (ARPE-19, ARPE-19 + shDJ-1, and hfRPE) were plated on MatTek Dishes, cultured, and infected with the various adenovirus as described above. Monolayers were incubated with 20 nM TMRE dissolved in cell culture medium for 15 min at 37 °C in a dark chamber. After incubation, monolayers were transferred in cell culture medium (1% serum) to confocal microscope for live cell imaging (excitation = 549 nm; emission = 575 nm). Images were acquired with similar settings. Quantification of TMRE fluorescence signal intensity was performed using ImageJ 2 as previously described [40]. Intensity signals were normalized to the signal of cells transduced with the empty vector (Ad).

### 4.11. Statistical Analysis

Data were analyzed using GraphPad Prism v8.1.1 (GraphPad Ssoftware, La Jolla, CA, USA) and are presented as the mean ± standard deviation (SD). Ordinary one-way ANOVA and two-way ANOVA with Tukey’s multiple comparisons test and unpaired, two-tailed Student’s *t*-test were used for determining statistical significance between groups with an alpha value of 0.05, and *p* values ≤ 0.0001 were considered statistically significant.

## Figures and Tables

**Figure 1 ijms-23-09938-f001:**
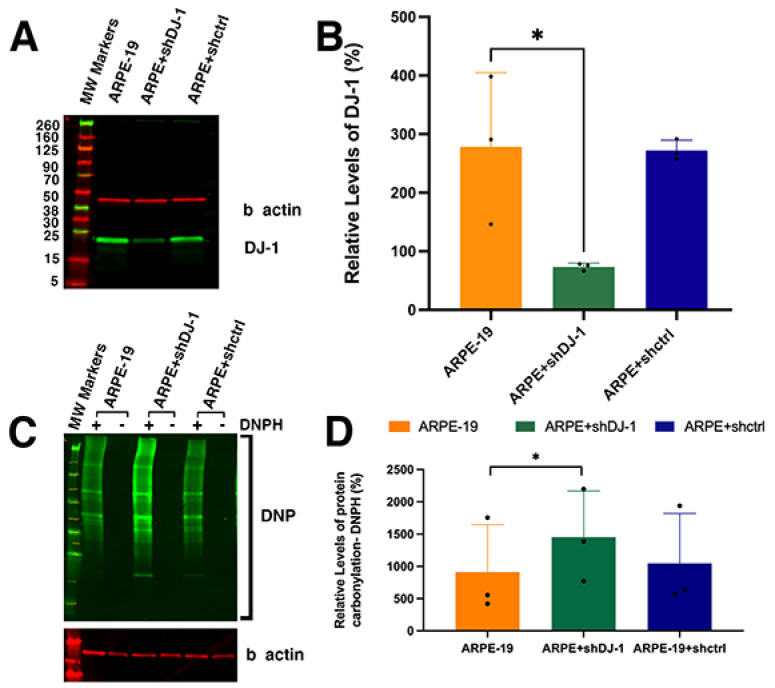
ROS is increased in RPE cells with DJ-1 downregulation. (**A**) Representative Western blot of RPE monolayers reacted with DJ-1 and β actin. (**B**) Total levels of DJ-1 protein in the RPE lysates. (**C**) Representative Western blot of RPE monolayers reacted with DNP (carbonyl groups) and β actin. (**D**) Total levels of DNP in the RPE lysates. Data are expressed as mean ± SD. * *p* = 0.01 to 0.05; one-way ANOVA; *n* = 3. Dots in (**B**,**D**) = individual experiments; asterisks above for significance.

**Figure 2 ijms-23-09938-f002:**
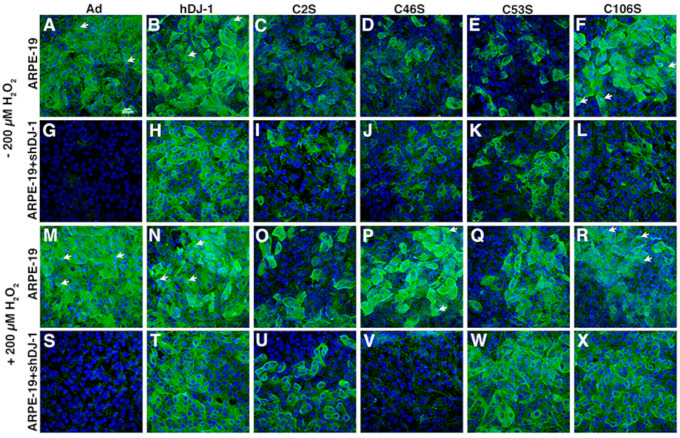
Under oxidative stress conditions, ARPE-19 monolayers transduced with the several DJ-1 constructs displayed visually higher immunoreactivity for DJ-1. (**A**–**F**) ARPE-19 and (**G**–**L**) ARPE-19 + shDJ-1 monolayers stained with DJ-1 (green) in baseline conditions. (**M**–**R**) ARPE-19 and (**S**–**X**) ARPE-19 + shDJ-1 monolayers stained with DJ-1 (green) under conditions of oxidative stress; blue: TO-PRO-3. Arrows = nuclear localization. Bar = 20 μm.

**Figure 3 ijms-23-09938-f003:**
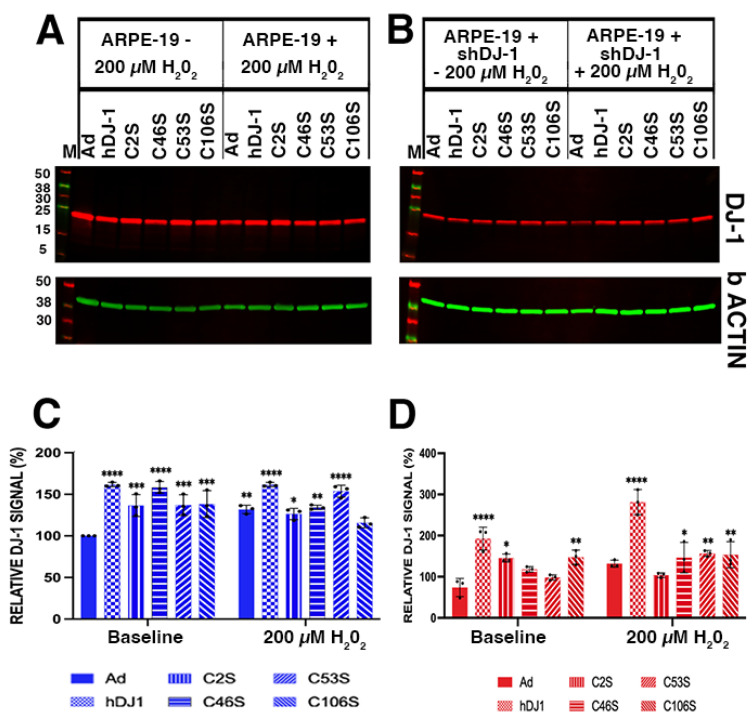
Under oxidative stress conditions, ARPE-19 monolayers transduced with the several DJ-1 constructs displayed significantly higher immunoreactivity for DJ-1. (**A**,**B**) Representative Western blot of RPE monolayers reacted with DJ-1 and b actin; (**A**) ARPE-19 (blue bars); (**B**) ARPE-19 + shDJ-1 (red bars). (**C**,**D**) Total levels of DJ-1 protein in the RPE lysates. (**C**) ARPE-19; (**D**) ARPE-19 + shDJ-1. Data are expressed as mean ± SD; levels were normalized to ARPE-19 + Ad. * *p* = 0.01 to 0.05; ** *p* = 0.001 to 0.01; *** *p* = 0.0001 to 0.001; **** *p* < 0.0001; two-way ANOVA; *n* = 3. Dots in (**C**,**D**) = individual experiments; asterisks above for significance.

**Figure 4 ijms-23-09938-f004:**
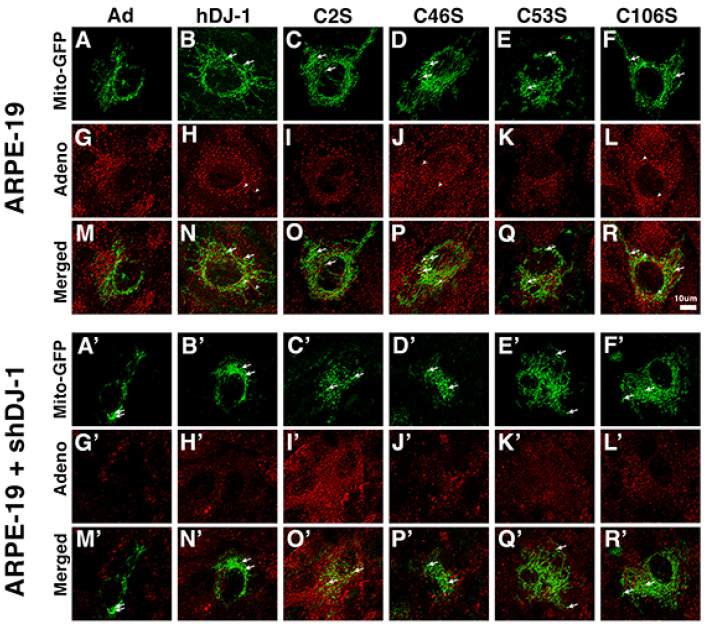
Transduction of several DJ-1 constructs in ARPE-19 monolayers under baseline conditions induces morphological changes in the mitochondria. (**A**–**R**) ARPE-19 and (**A’**–**R’**) ARPE-19 + shDJ-1 monolayers infected with CellLight^®^ mito-GFP (green, (**A**–**F** and **A’**–**F’**), transduced with the DJ-1 adenovirus constructs and stained with adenovirus type 5 (red) in baseline conditions; arrows = dilated tubules; arrowheads = adenovirus (exogenous DJ-1). Bar = 10 μm. Post-imaging processing using Lightning was conducted to generate average photon count to better show localization.

**Figure 5 ijms-23-09938-f005:**
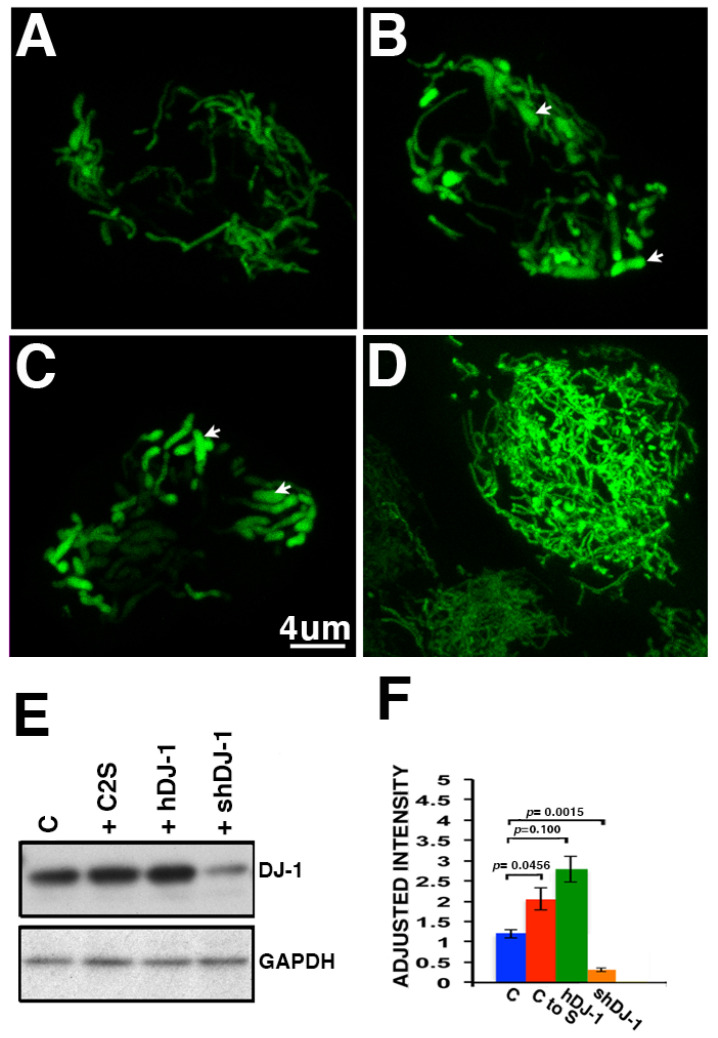
Transduction of hfRPE monolayers with the DJ-1 adenoviruses and lentiviruses and mitochondria morphology. Representative images of hfRPE monolayers transduced with (**A**) the control (Ad), (**B**) the human DJ-1 (hDJ-1), (**C**) human DJ-1 with the cysteine at residues 46, 53, and 106 mutated to serine (C2S), and (**D**) human shPARK followed by infection with CellLight^®^ Mito-GFP (green); arrows = dilated tubules. (**E**) Western blot analysis of lysates of the same cells reacted with DJ-1 and GAPDH antibodies. (**F**) Quantification of DJ-1 immunoreactivity ± SEM, *n* = 3. Bar = 4 μm.

**Figure 6 ijms-23-09938-f006:**
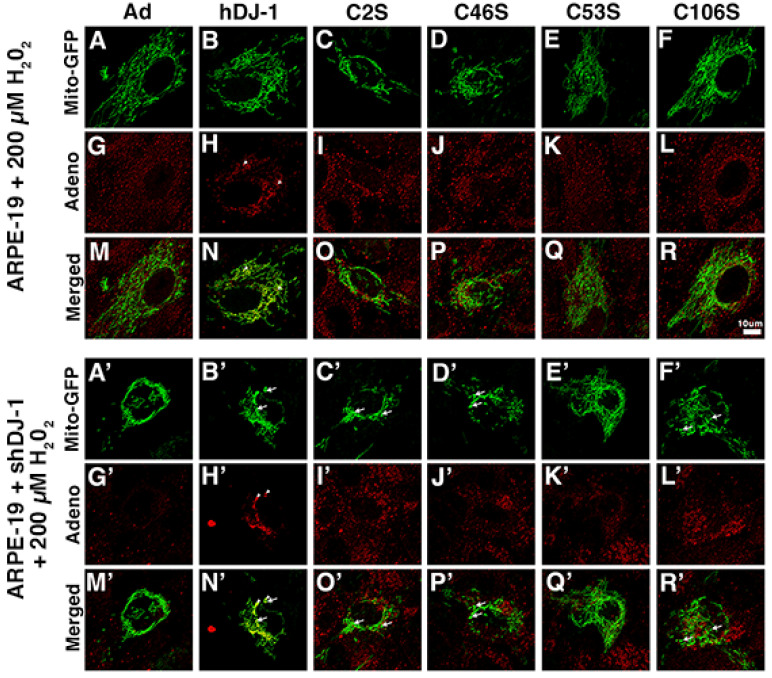
Transduction of several DJ-1 constructs in ARPE-19 + shDJ-1 monolayers under oxidative stress conditions induces mitochondria concentrated in the center of the RPE cells. (**A**–**R**) ARPE-19 and (**A’**–**R’**) ARPE-19 + shDJ-1 monolayers transduced with CellLight^®^ mito-GFP (green, **A**–**F** and **A’**–**F’**) and stained with adenovirus type 5 (red) in baseline conditions. Arrows = dilated tubules; arrowheads = adenovirus (exogenous DJ-1). Bar = 10 μm. Post-imaging processing using Lightning was conducted to generate average photon count to better show localization.

**Figure 7 ijms-23-09938-f007:**
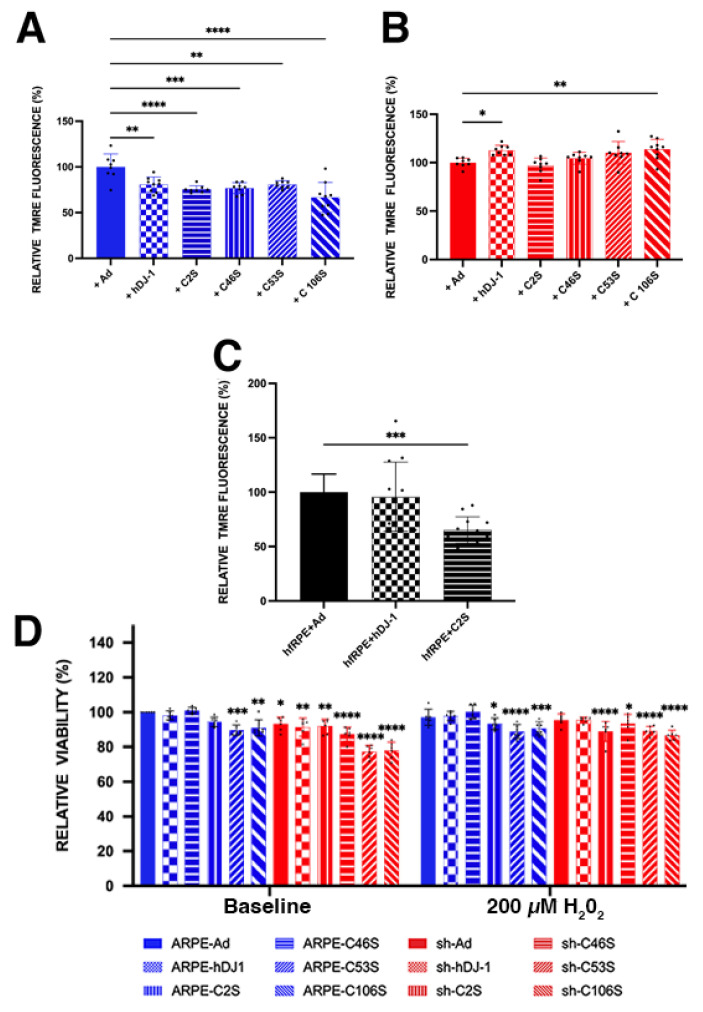
Transduction of ARPE-19 (blue bars), ARPE-19 + shDJ-1 (red bars), and hfRPE monolayers (black bars) with the several DJ-1 constructs displayed changes in mitochondrial membrane potential and viability. Representative TMRE fluorescence (**A**–**C**) and cell viability bar graphs (**D**). Data are expressed as mean ± SD; levels were normalized to ARPE-19 + Ad. * *p* = 0.01 to 0.05; ** *p* = 0.001 to 0.01; *** *p* = 0.0001 to 0.001; **** *p* < 0.0001; two-way ANOVA; *n* = 3. Dots inside graphs = Individual reading; asterisks above for significance.

**Table 1 ijms-23-09938-t001:** Summary of impact of DJ-1 expression changes and oxidation of its C residues on RPE function.

Cellular Localization and Function	shDJ-1	hDJ-1	C2S	C46S	C53S	C106S
ROS ^1^	increased	N/A ^2^	N/A ^2^	N/A ^2^	N/A ^2^	N/A ^2^
Detergent Solubility ^3^			increased	increased	increased	
Nuclei Localization		observed		observed		observed
Mitochondria Morphology	shorter and fragmented (ARPE-19), over-extended tubules (hfRPE)	dilated tubules (ARPE-19 & hfRPE)	dilated tubules (ARPE-19 & hfRPE)	dilated tubules	dilated tubules	dilated tubules
Mitochondria Network			very fragmented (ARPE-19 + shDJ-1)	more elaborated (ARPE-19 + shDJ-1)	fragmented (ARPE-19), highly elongated (ARPE-19 + shDJ-1)	highly elongated (ARPE-19 + shDJ-1)
Mitochondrial Co-localization		observed		observed		observed
ΔΨm ^4^		decreased (ARPE-19), increased (ARPE-19 + shDJ-1)	decreased (ARPE-19), decreased (hfRPE)	decreased (ARPE-19)	decreased (ARPE-19)	decreased (ARPE-19), increased (ARPE-19 + shDJ-1)
Viability					decreased	decreased

^1^ ROS: reactive oxygen species; ^2^ N/A: not assayed; ^3^ after triton X-100 incubation; ^4^ ΔΨm: mitochondrial membrane potential.

## Data Availability

The data presented in this study are available in the article and on request from the corresponding author.

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
