# Peer review of "Oxidation of DJ-1 Cysteines in Retinal Pigment Epithelium Function"

_ijms, 2022, doi:10.3390/ijms23179938_

Round 1

Reviewer 1 Report

Bhattacharya et al. found that RPE cells with reduced DJ-1 levels (ARPE-19+shDJ-1 cells) are more sensitive to low oxidative stress. Experiments on localization demonstrated that full-length DJ-1 and C106S overexpression boosts its distribution to the RPE mitochondria. Furthermore, they determined that oxidation of C53 is required for DJ-1 to exert its antioxidant function in RPE cells. These data indicate that DJ-1 undergoes unique PTMs in RPE cells. The study is intriguing and well-constructed, however certain adjustments are necessary: -

1. All of the figures are distorted and horizontally stretched, distorting the x-axis labeling.

2. One of the factors that cause oxidative stress is hypoxia. Did the authors investigate the effect of DJ1 knockdown and overexpression with three C residues in response to hypoxia?

3. How did the authors come to the choice to use only 200micromolar?

4. According to the authors' conclusion, DJ-1 cysteines modulate endogenous ROS levels. It will be fascinating to see how DJ-1 residues affect the levels of mt ROS.

Author Response

Bhattacharya et al. found that RPE cells with reduced DJ-1 levels (ARPE-19+shDJ-1 cells) are more sensitive to low oxidative stress. Experiments on localization demonstrated that full-length DJ-1 and C106S overexpression boosts its distribution to the RPE mitochondria. Furthermore, they determined that oxidation of C53 is required for DJ-1 to exert its antioxidant function in RPE cells. These data indicate that DJ-1 undergoes unique PTMs in RPE cells. The study is intriguing and well-constructed, however certain adjustments are necessary:

We thank the reviewer for the kind and encouraging assessment of our study.

  1. All of the figures are distorted and horizontally stretched, distorting the x-axis labeling.

Please read the response to comment 1 of reviewer #2. We apologize for the technical issue with the figures.

  1. One of the factors that cause oxidative stress is hypoxia. Did the authors investigate the effect of DJ1 knockdown and overexpression with three C residues in response to hypoxia?

The reviewer raises a fascinating topic since hypoxia increases ROS and leads to oxidative stress. Moreover, different types of stress may result in different stress modulation of DJ-1 (discussed in lanes 483-484). Therefore, future studies will analyze the role of DJ-1 C residues on different types of oxidative stress.

  1. How did the authors come to the choice to use only 200micromolar?

Please read the response to comment 2 of reviewer #2.

  1. According to the authors' conclusion, DJ-1 cysteines modulate endogenous ROS levels. It will be fascinating to see how DJ-1 residues affect the levels of mt ROS.

This is another fascinating topic raised by the reviewer. Future studies will analyze the role of DJ-1 C residues on the regulation of mtROS. This possibility was discussed in the revised text of the manuscript (lanes 503-505).

Reviewer 2 Report

This article is mainly focused on the roles of DJ1 on retinal epithelium cells. The finding of this research is quite interesting, however several issues needed to be addressed and improved, including, 

1.) The critical point for this manuscript is quality of figures. All of figures are difficult to interpret.  Authors have to modify all figures to an appropriate proportion with much greater resolution. 

2.) Hydrogen peroxide is quite easy to degrade. Single treatment with hydrogen peroxide for 17 h could not be a good representative for chronic oxidative stress. Authors have to explain the rational of experimental condition.

3.) Authors also have to describe why hydrogen peroxide at 200 micromolar is a model of "low oxidative stress". What is the criteria for low v.s. high oxidative stress? Does hydrogen peroxide at this concentration causes any toxicity to retinal epithelium cells? 

4.) There are inconsistency of information on measurement of protein carbonyl formation between material&methods v.s. results section. On material&methods, authors stated that hydrogen peroxide was used as an inducer for oxidative damage; while using of hydrogen peroxide was not addressed and discussed on results section (Figure 1C). H2O2 is a key factor for the observation of oxidative protein damage.   

5.) Determination of oxidation of protein is the only experiment that observe effects of DJ1 on oxidative stress. Additional experiments, e.g. measurement of redox biomarkers, or evaluation of cellular oxidative status, are required to support these finding.

6.) In order to support roles of DJ1 on Nrf2 cascade and its antioxidant activities against oxidative stress, the observation on downstream targets of Nrf2 by using western blot analysis or RT-QPCR are recommended.  

7.) Due to poor quality of figures, it is very difficult to interpret data from confocal microscope. Authors have to improve on explanation of  morphological change, such as diffuse cytoplasmic pattern or alteration in mitochondrial structure. It is very difficult for audience to follow and  understand.   

8.) On Figure 3B, there are strong non-specific signal on hDJ-1 (green fluorescent smear). Hence, the significant increase in DJ1 on Figure 3D could be a misinterpretation due to this non-specific problem. The authors have to clarify and discuss on the possibility of this issue.   

10.) What is a rational for counter-staining of adenovirus type 5 (red) on Figure 4?

11.) The authors demonstrated changes in morphology of mitochondria following transduction with DJ-1 construct. However, the authors did not observe whether these transductions cause any alteration in mitochondial function, or not. The observation on function of mitochondria will allow us to better understand contributions of DJ-1 against oxidative stress. 

11.)  In order to demonstrate a significant function of DJ1, the results from hfRPE studies should be included on main manuscript, rather than on supplementary section.  

Author Response

1.) The critical point for this manuscript is quality of figures. All of figures are difficult to interpret.  Authors have to modify all figures to an appropriate proportion with much greater resolution. 

All figures were prepared using AdobePhotoshop with a 300 dpi resolution. We added the figures to the ijms doc template. The original template looked fine, but we did not check the quality of the final version after uploading it to the journal’s site. For the revision, figure panels were uploaded again, and the quality of the figures in the final draft was confirmed.

2.) Hydrogen peroxide is quite easy to degrade. Single treatment with hydrogen peroxide for 17 h could not be a good representative for chronic oxidative stress. Authors have to explain the rational of experimental condition.

We agree with the reviewer about the degradation of H202. However, our experiments were designed based on previous experiments carried out in our lab had reported a dose-response relating DJ-1 and oxidized DJ-1 intensity in RPE cultures when cells were exposed for 1 and 18 hrs to increasing concentrations of H2O2 (Shadrach KG, Rayborn ME, Hollyfield JG, Bonilha VL (2013) DJ-1-Dependent Regulation of Oxidative Stress in the Retinal Pigment Epithelium (RPE). PLoS ONE 8(7): e67983. https://doi.org/10.1371/journal.pone.0067983.). Those experiments also identified a visible increase in immunocytochemical staining for DJ-1 under these conditions and established that overnight incubation with 400-1000 µM H2O2 resulted in significant death of RPE cells. This information was included in the text of the manuscript (lanes 189-195).

 3.) Authors also have to describe why hydrogen peroxide at 200 micromolar is a model of "low oxidative stress". What is the criteria for low v.s. high oxidative stress? Does hydrogen peroxide at this concentration causes any toxicity to retinal epithelium cells? 

Please read the response to question 2.

4.) There are inconsistency of information on measurement of protein carbonyl formation between material&methods v.s. results section. On material&methods, authors stated that hydrogen peroxide was used as an inducer for oxidative damage; while using of hydrogen peroxide was not addressed and discussed on results section (Figure 1C). H2O2 is a key factor for the observation of oxidative protein damage. 

The materials and methods section was corrected as requested.  

5.) Determination of oxidation of protein is the only experiment that observe effects of DJ1 on oxidative stress. Additional experiments, e.g. measurement of redox biomarkers, or evaluation of cellular oxidative status, are required to support these finding.

We have also analyzed the mitochondrial membrane potential (ΔΨm) by incubating the monolayers with tetramethylrhodamine ethyl ester (TMRE) and quantifying the signal. Results were compiled in a new figure (Figure 7). 

6.) In order to support roles of DJ1 on Nrf2 cascade and its antioxidant activities against oxidative stress, the observation on downstream targets of Nrf2 by using western blot analysis or RT-QPCR are recommended.  

Previous reports have shown that DJ-1 regulates Nrf2 in different cells. Moreover, our previous control and DJ-1 KO retinas and RPE analysis detected no significant changes in NRF2 levels in labeled cryosections. We also found that the baseline level of NRF2 genes was downregulated in the retinas but upregulated in the RPE of 3-month-old DJ-1 KO mice compared with age-matched WT mice. We agree with the reviewer about the need to better understand the role of DJ-1 cysteine oxidation in the RPE function. However, to avoid delaying the publication of this manuscript, we removed the Nrf2 panel figure from Figure 1 and revised the text accordingly.

7.) Due to poor quality of figures, it is very difficult to interpret data from confocal microscope. Authors have to improve on explanation of  morphological change, such as diffuse cytoplasmic pattern or alteration in mitochondrial structure. It is very difficult for audience to follow and  understand.   

Please read the response to comment 1.

8.) On Figure 3B, there are strong non-specific signal on hDJ-1 (green fluorescent smear). Hence, the significant increase in DJ1 on Figure 3D could be a misinterpretation due to this non-specific problem. The authors have to clarify and discuss on the possibility of this issue.   

The figure was replaced by a new set of western blot figures. The quantification was performed in tiny areas around the band signal; the graph presented is an average of three experiments.

9.) What is a rational for counter-staining of adenovirus type 5 (red) on Figure 4?

Monolayers reacted with the adenovirus type 5 antibody because the DJ-1 constructs were not tagged; this was an indirect way to ensure we were imaging cells expressing the exogenous DJ-1. This information was added to the text (lanes 256-257).

10.) The authors demonstrated changes in morphology of mitochondria following transduction with DJ-1 construct. However, the authors did not observe whether these transductions cause any alteration in mitochondial function, or not. The observation on function of mitochondria will allow us to better understand contributions of DJ-1 against oxidative stress. 

We agree with the reviewer's comment and have performed mitochondrial membrane potential (ΔΨm) labeling and measurements; please read the answer to comment 5.

11.)  In order to demonstrate a significant function of DJ1, the results from hfRPE studies should be included on main manuscript, rather than on supplementary section.

As requested, this figure was added to the manuscript (Figure 5). In addition, additional results of experiments quantifying mitochondrial membrane potential of hfRPE monolayers were added to the Figure 7.

Round 2

Reviewer 2 Report

The authors have well explained on previous round of questions. On this revised manuscript, there are only few points that the authors have to consider. 

1.) Figure 1C, there are "+" and "-" symbol. It is very difficult to understand the experimental condition. Adding DNPH on the side would be more appropriate. 

2.) Figure 3B, there are two bands on the DJ1 data. One would be DJ1, and other would be non-specific band. The authors have to specifically label the band of DJ1. Another recommendation is showing only DJ1 band. 

3.) The data on confocal microscopy (Figure 2) is not consistent to western blot (Figure 3). For instance, 

- Figure 2: C2S (2C), C46S (2D) and C53S (2E) has much lower DJ1 levels than Ad (2A): Lower green intensity v.s. Figure 3: C2S, C46S and C53S has greater DJ1 levels than Ad.

Figure 2: C2S (2U), C53S (2W) and C106S (2X) has much higher DJ1 levels than Ad (2S): Much brighter green color v.s. Figure 3: C2S has lower DJ1 levels than Ad. C53S and C106S has similar level to Ad. 

The author have to further discuss on this issue. 

4.) According to the results, each cysteine residue (C2, C46, C53, C106) of DJ1 has unique function. The authors have to summarize the findings from these studies and further discuss on the contribution of each cysteine residue on mitochondrial morphology, mitochondrial function and DJ1 expression/localization. 

Author Response

Response to Reviewer #2:

The authors have well explained on previous round of questions. On this revised manuscript, there are only few points that the authors have to consider. 

1.) Figure 1C, there are "+" and "-" symbol. It is very difficult to understand the experimental condition. Adding DNPH on the side would be more appropriate.

The figure was edited as requested.  

2.) Figure 3B, there are two bands on the DJ1 data. One would be DJ1, and other would be non-specific band. The authors have to specifically label the band of DJ1. Another recommendation is showing only DJ1 band. 

The figure was edited as requested.  

3.) The data on confocal microscopy (Figure 2) is not consistent to western blot (Figure 3). For instance, 

Figure 2: C2S (2C), C46S (2D) and C53S (2E) has much lower DJ1 levels than Ad (2A): Lower green intensity v.s. Figure 3: C2S, C46S and C53S has greater DJ1 levels than Ad.

Figure 2: C2S (2U), C53S (2W) and C106S (2X) has much higher DJ1 levels than Ad (2S): Much brighter green color v.s. Figure 3: C2S has lower DJ1 levels than Ad. C53S and C106S has similar level to Ad. 

The author have to further discuss on this issue.

The potential inconsistency between the confocal microscopy and western blot analysis was discussed in the text of the manuscript as requested (lanes 251-273).

4.) According to the results, each cysteine residue (C2, C46, C53, C106) of DJ1 has unique function. The authors have to summarize the findings from these studies and further discuss on the contribution of each cysteine residue on mitochondrial morphology, mitochondrial function and DJ1 expression/localization. 

We have added a table to the text of the manuscript to summarize the findings as requested.
